# Effect of Block-Based Python Programming Environment on Programming Learning

Yongcheon Kim [1], Jamee Kim [2] and Wongyu Lee [3,*]

1   Department of Computer Science Education, Graduate School, Korea University, Seoul 02841,
    Republic of Korea; kimyc3766@gmail.com
2   Major of Computer Science Education, Graduate School of Education, Korea University, Seoul 02841,
    Republic of Korea; celine@korea.ac.kr
3   Department of Computer Science and Engineering, Graduate School, Korea University, Seoul 02841,
    Republic of Korea
*   Correspondence: lee@inc.korea.ac.kr; Tel.: +82-2-3290-2391

**Abstract:** The advancement of computing technology has led to many changes in a variety of fields, and the importance of programming education has been emphasized in many countries worldwide. Despite the importance of programming education, the cognitive burden of text programming for beginners has not been reduced. The goal of this study was to implement an environment where a text programming language is used in a block-based programming environment and to determine at which school level this learning environment affects positive perceptions of programming. To achieve this goal, we conducted programming classes targeting 128 middle school, high school, and university students for 14 weeks and analyzed the effects of the factors of "understanding of programming instructions", "usage confidence", and "usefulness" on "positive perceptions of programming". The results of the analysis by school level show that "usefulness" influenced positivity toward programming for middle school students, "usefulness" and "understanding of programming instruction" for high school students, and "understanding of programming instruction" and "usage confidence" for university students. Therefore, the significance of this study confirms the need to construct the learning environment differently depending on school level, even for beginners.

**Keywords:** block-based python programming; programming environment; programming learning

## 1. Introduction

Beginning with the United Kingdom in 2013, the educational curricula of various countries, such as India, Korea, Japan, and Finland, have been revised to include software and artificial intelligence (AI) content [1,2]. Given the direct correlation between software and AI technology development, personnel training, and national competitiveness, countries worldwide are emphasizing the importance of programming education. The status of software and AI in each country can be found on the global AI index provided by Tortoise Media [3].

The importance of programming education began when computers were introduced to schools in the 1980s, when text-based programming languages, such as Basic and Logo, were initially taught [4]. However, the difficulty of the concepts and syntax that beginners had to learn in these text-based programming courses was a factor that raised a barrier to entry [5,6]. The use of complex programming languages with variables, loops, arrays, functions, etc. was a major reason for beginners to fail at programming [7]. Inputting commands accurately is essential in a text-based programming language, but it is often perceived as difficult for beginners [8,9]. For example, text input for programming is a major source of syntax errors, and the use of symbols and punctuation marks that beginners do not recognize is frustrating for them [10,11]. Even when receiving error messages, it is

difficult to find and correct errors without recognition of commands, punctuation, symbols, etc. and accurate memorization of grammar [12–15].

Research on teaching methods and programming environments has been conducted to solve the difficulties that beginners experience in the programming process [16]. As error feedback is one of the causes of difficulties for beginners in the programming process [17], Scratch, which is learned through trial and error, has been used to educate beginners regardless of school level [18]. University students have also shown high satisfaction with their success in programming [19]. However, block-based programming has a limitation in that it is disconnected from text programming languages used in industry, which increases the difficulty of transitioning to text-based programming [4,20]. Hybrid-based programming environments, such as Blockly [21] and Pencil Code [22], in which blocks are converted into text commands when combined, have been developed, but they have not reduced the burden of learning text commands. As there is a difficulty in the transition of concepts learned from block-based into text-based programming, MakeCode for micro:bit has also been used to perform "real" text-based programming [23].

To lower the barriers to programming, the advantages of a block-based environment need to coexist with the sense of accomplishment that comes from text-based programming. This is because a sense of efficacy in learning should have a positive impact on motivation for programming [24,25]. Accordingly, various hybrid-based programming studies have been conducted. The effectiveness of Parsons Problems, whereby text commands are composed using blocks, has been demonstrated [26,27], and Pencil.cc [28], with which the effectiveness of classes for high school students has been proven, utilizes Pencil Code and has been reported to be suitable for university students [29,30]. In another study, Scratch and Blockly were compared for female students aged 10 to 14 [21]. These studies on hybrid-based programming environments are limited to specific school levels and do not specify which factors influence the positive perceptions of programming. Thus, it is necessary to show which factors contribute to a positive change in perceptions of programming rather than the importance of constructing the environment itself.

Therefore, this study aimed to provide a text-based programming language in a block-based programming environment and to identify the factors of the learning environment that affect the positive perceptions of programming for beginners at different school levels. To achieve the objectives of this study, we chose Python 3.9.0 as a text programming language for the construction of a hybrid learning environment. Python had a global programming language share of 13.58% in the TIOBE index as of January 2022, which is higher than that of the C (12.44%) and Java (10.66%) languages. From the perspective of constructing a programming environment, Python has the advantage of compiling the code line-by-line, so if an error occurs, beginners can accurately identify the location of the error. Furthermore, Python is a highly useful language as it is a supported language of TensorFlow, which is related to artificial intelligence and big data analysis, and PyTorch uses Python as a scripting language [31,32].

Proceeding with text-based programming in a block-based programming environment by school level and identifying factors that influence positive perceptions of programming suggest that learning points should be different for different school levels, even in the same learning environment. While various studies on programming beginners have only identified the limits of difficulty in programming, this study aimed to provide direction for introductory programming classes that should be structured differently for different school levels.

## 2. Related Work

### 2.1. Factor Analysis

When developing a programming environment and evaluating its usability, the validity of the evaluation tools must be ensured. In other words, it is necessary to verify that the tools developed to evaluate usability contain questions that are suitable for evaluation. Validity refers to whether the measurement target can be measured appropriately [33].

Additionally, to evaluate two or more pieces of content, it is important to confirm whether the corresponding constructs can be used to evaluate what they are meant to evaluate. To confirm the validity of a construct, factor analysis can be used, whose purpose is to reveal the covariance structure of the data variables. It is used to create a single construct when one variable changes with another. The process for confirming the validity of the usability evaluation tool and performing factor analysis to create the constructs is as follows:

(1)    Create questions, perform a usability analysis, and obtain scores for each evaluation question.
(2)    Calculate a matrix of the correlation coefficients between questions.
(3)    Extract nonrotated factors.
(4)    Rotate the factors.
(5)    Interpret and assign names based on the content of questions with high factor loadings related to rotated factors.

Factor rotation is used to obtain a structure wherein the variables of each factor can be clearly interpreted. The goal is to consider factors that are not accurately explained through factor loading, which indicates the degree to which each variable reflects a single factor, and convert them into a simple structure. By rotating to a simple structure, each variable receives a high load from only one factor and relatively low loads from others, simplifying the factor structure. Thereafter, it can be interpreted as a factor structure that is not explained by the initial factor load. The following factor equation can be used to derive factors $F_1$, $F_2$, ..., $F_k$ which include the weight coefficients $a_1$, $a_2$, ..., $a_k$ of multiple variables in the data:

$$Z_j = a_{j1}F_1 + a_{j2}F_2 + \ldots + a_{jk}F_k + U_j$$

$Z_j$ = Standard score of the $j_{th}$ variable;
$a_{jk}$ = Weight (coefficient) for factor $k(F_k)$ of the $j_{th}$ variable;
$U_j$ = Unique variance of the $j_{th}$ variable.

Factor analysis was used to extract constructs for measuring their effectiveness and satisfaction toward education and to ensure their validity. Lee (2019) developed 48 evaluation items to measure the relationship between motivation and achievement based on student engagement in an e-learning environment. Furthermore, six constructs (psychological motivation, peer collaboration, cognitive problem solving, interaction with instructors, community support, and learning management) were extracted to conduct the study [34]. To determine why beginner programmers have low coding skill levels, factor analysis was conducted on the answers to programming problems submitted by 614 university students through a web-based learning system, and four skill-level constructs were extracted: code style, syntactic, logical error-related, and syntax debugging [35]. Factor analysis is a method used to extract underlying constructs from a set of observed variables, categorizing confirmed constructs based on shared variance and extracting content commonly explained by multiple evaluation questions.

Thus, factor analysis ensures the validity of the developed tools and can be utilized to extract factors that commonly explain the questions within an examination tool.

### 2.2. Regression Analysis

Regression analysis is a statistical technique that enables the prediction of values for dependent variables based on the values of independent variables [36]. It accomplishes this by establishing linear equations that represent the relationships between the independent and dependent variables. In other words, it examines the extent to which dependent variables change based on the changes in independent variables and estimates the predictive power of independent variables with regard to dependent variables. Simple regression analysis assumes linear relationships between independent and dependent variables and is expressed as follows:

$$Y' = \beta_0 + \beta_1 X_i + \varepsilon_i$$

$\beta_0$: When $X_i = 0$, the expected value of $Y_i$ (regression constant, intercept);

$\beta_1$: Population's regression coefficient (slope of regression line);
$\varepsilon_i$ : Error that is not explained by $X_i$.

In regression analysis, the least squares method is utilized to estimate the intercept ($\beta_0$) and regression ($\beta_1$) coefficients. These coefficients minimize differences between the actual values of the dependent variable ($Y$) and the predicted values ($Y'$) obtained using the independent variables. The goal is to find the regression equation that minimizes the overall difference between the observed and predicted values of the dependent variable.

The total change in the $Y$ value is classified into two parts: those that can and cannot be explained by the regression equation.

$$\sum(Y_i - \overline{Y})^2 = \sum(Y_i' - \overline{Y})^2 + \sum(Y_i - Y')^2$$

$$SS_T = SS_R + SS_E$$

$SS_T$ : Total deviation sum of squares of $Y$;
$SS_R$ : Change that can be explained by the regression equation (regression deviation sum of squares);
$SS_E$ : Change that cannot be explained by the regression equation (residual sum of squares).

An analysis of the variance table for simple regression can be seen in Table 1.

**Table 1.** Block-based programming environment analysis results.

| | Sum of Squares ($SS$) | df | Mean Square (MS) | F | $R^2$ |
|---|---|---|---|---|---|
| Regression | $SS_R$ | 1 | $SS_R/1$ | $MS_R/MS_E$ | $SS_R/SS_T$ |
| Residual | $SS_E$ | $n-2$ | $SS_R/n-2$ | | |
| Total | $SS_T$ | $n-1$ | | | |

One criterion for judging the suitability of a regression equation is the determination coefficient ($R^2$). It indicates the explanatory power of an independent variable for a dependent variable and refers to the ratio of the variance, obtained using the regression equation, to the total variance of the dependent variable. The closer the value of the determination coefficient is to 1, the greater the explanatory power of the independent variable [37].

$$R^2 = \frac{SS_R}{SS_T} = 1 - \frac{SS_E}{SS_T}$$

Simple regression analysis is used for analyzing the predictive power of an independent variable with regard to a dependent variable, whereas multiple regression analysis is a statistical method used to determine the variable that affects the dependent variable among several independent variables. The linear equation for the multiple regression model is as follows:

$$Y_i = \beta_0 + \beta_1 X_{1i} + \beta_2 X_{2i} + \ldots + \beta_k X_{ki} + \varepsilon_i$$

where $\beta_k$ is the unstandardized regression coefficient, i.e., the partial slope of the regression equation. It indicates the change in the $Y$ value when the value of a certain independent variable $X_k$ is increased by 1, while those of the other independent variables are fixed.

There are several methods for selecting the variables that must be included in the regression model to find the optimal regression equation, such as enter, forward selection, backward elimination, and stepwise selection [18]. Stepwise selection determines the optimal regression equation through an appropriate combination of adding and removing independent variables. As the variables are added individually, the significance of the variables already included in the model is reviewed, and those that are insignificant are excluded. This method is helpful for extracting significant variables.

Multiple regression analysis has been used to determine the factors that affect students' attitudes toward computer programming, which are a sense of achievement during the programming process, self-efficacy with regard to programming, and recognition learning [38]. Additionally, studies have been conducted to predict students' levels of academic achievement in the early stages [39]. In some cases, path diagrams have been used to visualize the extent to which two or more explanatory variables affect a dependent variable [40]. This is because path diagrams of the effects of three independent variables ($X$, $U_1$, and $U_2$) on a dependent variable ($Y$) can help users understand the relationships between them and interpret their significance. Figure 1 shows a path diagram in which mindfulness-based relapse prevention (MBRP) treatment, treatment period (CRAVE0), and hours in treatment (TREATHRS) are designated as independent variables to identify factors that affect alcoholism treatment.

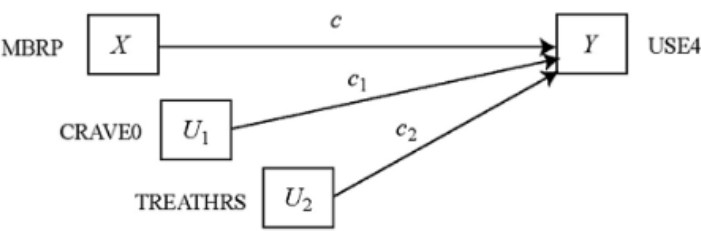

**Figure 1.** A path diagram of multiple linear regression.

## 3. Programming Environments

Figure 2 shows the components of the programming environment used for data collection.

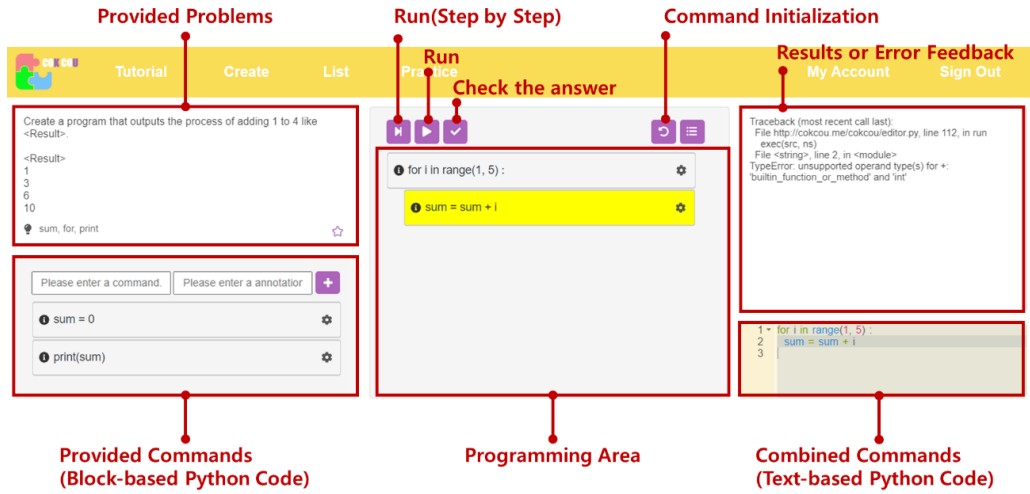

**Figure 2.** Programming activity user interface (UI).

First, "Provided Problems" area. Learners were provided with some problems and asked to develop programs based on their content. For example, if the problem was "Print the process of adding numbers 1–4", then "1, 3, 6, 10" must be printed.

Second, "Provided Commands" area. In some cases, all commands required to solve the problem were provided, and in others, too many or too few commands were provided. If too few commands were provided, the learner was required to use the "+" button to add commands directly. For the provided commands, a single-line Python program command constituted a single block. The learner could move commands by dragging and dropping them using the mouse instead of entering the text.

The third component was the "Programming" area, wherein the learners dragged and dropped commands from the provided command area to complete the program.

Unnecessary commands could either be deleted or modified by clicking the Modify button (✿). The Run (▶) button could be pressed to execute the combined commands and see the results. The Run One Step (▶I) button could be used to run a command one line at a time, obtain the results, and view the locations of commands where errors have occurred during the debugging process, which is useful. Clicking the Run or Run One Step buttons showed commands causing errors in yellow color. To initialize with the first command given, the Command Initialization (↻) button must be clicked.

The fourth component was an area for viewing "the execution results and error feedback". If there were no errors, the execution results of the combined commands were output. In case of errors, the programming environment outputs the locations of the commands where errors occurred and the causes of these errors.

Finally, the fifth component was an area to enter text-based Python commands. When block-based Python commands are combined in the programming area, the content of the "Combined commands area" changes.

Figure 3 shows examples of programs created by a student using COKCOU (COding from Kindergarten COding to University). The first example is a program for distinguishing between uppercase and lowercase letters in ASCII code. This program uses a combination of conditional statements to determine whether the entered alphabet letter is uppercase or lowercase. It provides the indentation functionality along with variables, operators, conditional statements, and output statements to help beginners perform Python programming.

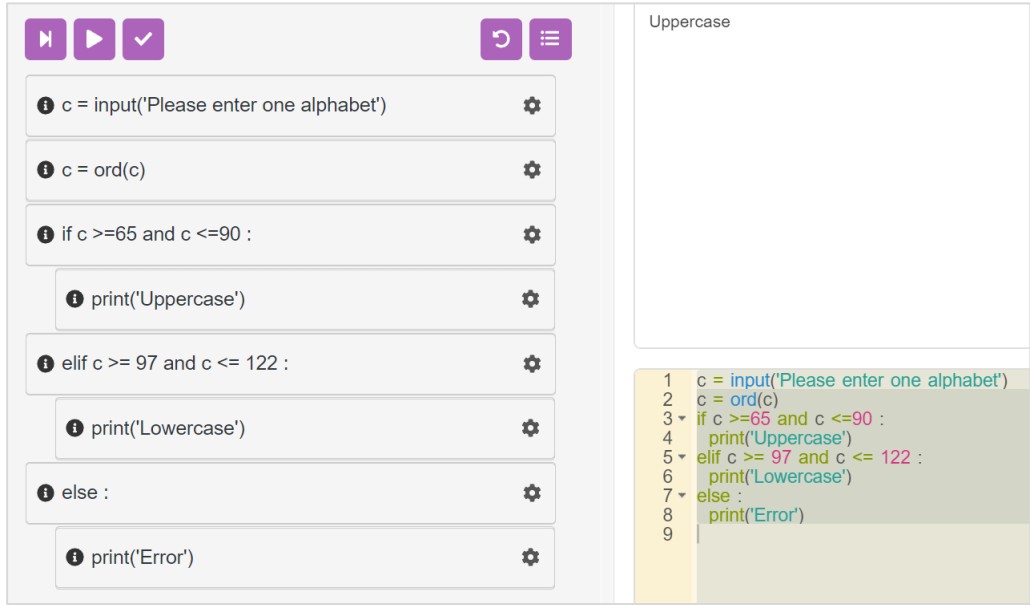

**Figure 3.** Program for distinguishing between uppercase and lowercase letters in ASCII code.

Figure 4 shows example is a program that finds the largest of five numbers entered. One of the functionalities that beginners find difficult in Python programming is indentation. COKCOU provides a constant indentation size to help reduce errors caused by indentation.

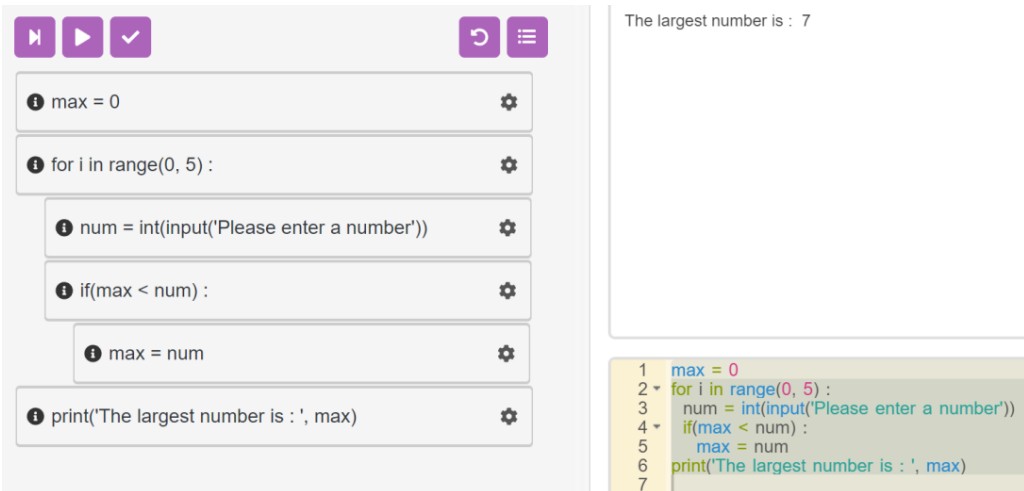

**Figure 4.** Program that finds the largest of five numbers entered.

## 4. Methods

### 4.1. Participants

To determine the effects of the proposed programming environment on beginners' positive perceptions of programming, one semester of Python classes was conducted with 22 middle school, 34 high school, and 72 university students who had never used Python.

### 4.2. Programming Course

Additionally, to determine whether a block-based Python programming environment is best suited for middle school, high school, or university students, we conducted the study procedure shown in Figure 5.

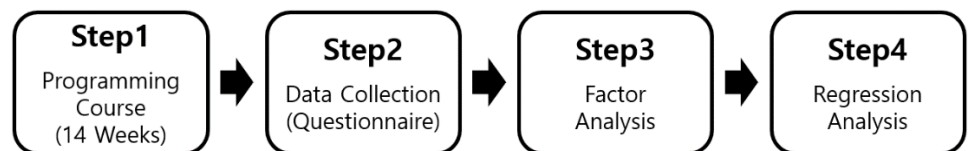

**Figure 5.** Study procedure.

In Step 1, a programming course was provided to middle school, high school, and university students through a 14-week programming course. In Step 2, we conducted a survey for each group after the class. In Step 3, we conducted a factor analysis to ensure the construct validity of the questionnaire. The construct validity aimed to determine whether the content of the questionnaire is appropriate for the content to be measured (see Section 2.1). In Step 4, we performed a multiple regression analysis to determine the factors that affect positivity toward the programming course by school level (see Section 2.2).

#### 4.2.1. Procedure

The classes were held for 2 h per week for 14 weeks, and the class content included output (print), input, operations (the four arithmetic operations, logical operations, and comparison operations), conditions (if, elif, and else), iteration (for, while), lists, Python's built-in functions, user-defined functions, and individual projects.

#### 4.2.2. Data Collection

Table 2 presents the content of the survey regarding the "programming learning environment" and "positive perceptions of programming" answered by the 128 participants. The survey used a 5-point Likert scale, where 5 = "strongly agree" and 1 = "strongly disagree". The survey was conducted after the classes were finished.

**Table 2.** Questionnaire.

| Category | No. | Item |
|---|---|---|
| Programming Learning Environment | A01 | I understand commands. |
| | A02 | Commands are easy to use. |
| | A03 | I am confident in using commands. |
| | A04 | I have the knowledge and techniques required for using commands. |
| | A05 | I can obtain the desired results. |
| | A06 | It helps to understand "print( )" |
| | A07 | It helps to understand "input( )" |
| | A08 | It helps to understand "quadratic/comparative/logical operations" |
| | A09 | It helps to understand "if, elif, else" |
| | A10 | It helps to understand "for, while" |
| | A11 | It helps to understand "list" |
| | A12 | It helps to understand "function" |
| | A13 | It helps to understand "algorithm" |
| | A14 | The environment helps with my programming activities. |
| | A15 | I want to spend more time using the provided environment. |
| | A16 | I want to use the provided environment in the future. |
| Positive Perceptions of Programming | B01 | Programming helps create a better world. |
| | B02 | Programming is worth studying. |
| | B03 | Programming will be useful even after I graduate school. |
| | B04 | Programming is relevant to the environment, technology, and society. |
| | B05 | The programming class hours at school should be increased. |
| | B06 | Programmers think and make decisions rationally. |
| | B07 | I want to know more about programming. |

The "Programming Learning Environment" was constructed based on the studies by Chuang (2020) [41], Cheng (2019) [42], Moons (2013) [43], and Zorn (2013) [44]. "Positive Perceptions of Programming" were constructed based on the studies by Tang (2020) [45], Kong (2020) [46], Kong (2018) [47], Alothman (2017) [48], and Rubio (2015) [49].

4.2.3. Factor Analysis

To determine the factors of the programming environment that affected beginners' "positive perceptions of programming" in this study, factor analysis was conducted through the following steps: data suitability assessment, factor extraction, and factor rotation. SPSS (version 26.0; IBM Corp., Armonk, NY, USA) for Windows was used for factor analysis.

First, to verify the suitability of the factor analysis model, the Kaiser–Meyer–Olkin (KMO) test and Bartlett's test of sphericity were performed on the survey content "programming learning environment" as presented in Table 3. The KMO value was 0.924, and Bartlett's test of sphericity significance probability was 0.000. Thus, the factor analysis model was suitable.

**Table 3.** Kaiser–Meyer–Olkin and Bartlett's test of sphericity.

| Kaiser–Meyer–Olkin (KMO) Measure of Sampling Adequacy | | 0.924 |
|---|---|---|
| **Bartlett's Test of Sphericity** | **Approx. Chi-Square** | 3475.178 |
| | **df** | 253 |
| | **Sig.** | 0.000 |

Second, the factors were extracted. Principal component analysis and exploratory factor analysis were used in this study. Table 4 presents the total explained variance. There were 23 components before extraction; however, after extraction and rotation, there were 4 components with eigenvalues greater than 1. Apparently, the four extracted factors accounted for 81.06% of the total variance and could be considered to comprise high explanatory power.

**Table 4.** Total variance explained.

| | Initial Eigenvalues | | | Rotation Sums of Squared Loadings | | |
|---|---|---|---|---|---|---|
| Component | Total | % of Variance | Cum. % | Total | % of Variance | Cum. % |
| 1 | 13.727 | 59.684 | 59.684 | 6.496 | 28.243 | 28.243 |
| 2 | 2.223 | 9.664 | 69.348 | 5.772 | 25.095 | 53.338 |
| 3 | 1.684 | 7.324 | 76.671 | 3.546 | 15.418 | 68.756 |
| 4 | 1.010 | 4.389 | 81.060 | 2.830 | 12.305 | 81.060 |

Third, a factor rotation was performed. This study used an orthogonal rotation method based on varimax, which uses Kaiser normalization. Table 5 shows the rotated component matrix. The following Table 5 presents the results of reducing factors using the rotated component matrix.

The model settings were validated through dimensionality reduction. The model used in this study is Figure 6.

The following Table 6 shows the averages and standard deviations for each grade level, according to this proposed model.

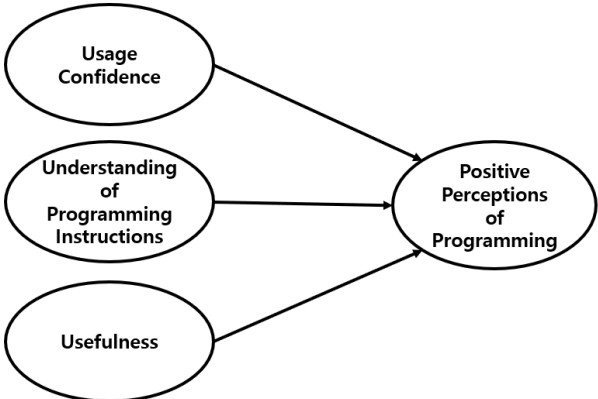

**Figure 6.** Path diagram of the impact of programming learning environment factors on "Positive Perceptions of Programming".

**Table 5.** Rotated factor matrix.

| Category | Subcategory | No. | Item | Factor | | | |
|---|---|---|---|---|---|---|---|
| Programming Learning Environment | Understanding of Programming Instructions | A11 | It helps to understand "list" | 0.847 | 0.263 | 0.138 | 0.285 |
| | | A08 | It helps to understand "quadratic/comparative/logical operations" | 0.845 | 0.203 | 0.258 | 0.314 |
| | | A09 | It helps to understand "if, elif, else" | 0.835 | 0.291 | 0.213 | 0.285 |
| | | A13 | It helps to understand "algorithm" | 0.818 | 0.227 | 0.189 | 0.276 |
| | | A10 | It helps to understand "for, while" | 0.792 | 0.338 | 0.093 | 0.306 |
| | | A07 | It helps to understand "input( )" | 0.731 | 0.290 | 0.237 | 0.356 |
| | | A12 | It helps to understand "function" | 0.702 | 0.465 | 0.109 | 0.243 |
| | | A06 | It helps to understand "print( )" | 0.627 | 0.438 | 0.248 | 0.352 |
| | Usage Confidence | A03 | I am confident in using commands. | 0.352 | 0.822 | 0.002 | 0.236 |
| | | A05 | I can obtain the desired results. | 0.363 | 0.705 | 0.239 | 0.239 |
| | | A04 | I have the knowledge and techniques required for using commands. | 0.513 | 0.675 | 0.068 | 0.293 |
| | | A01 | I understand commands. | 0.467 | 0.626 | 0.208 | 0.335 |
| | | A02 | Commands are easy to use. | 0.232 | 0.594 | 0.531 | 0.204 |
| | Usefulness | A16 | I want to use the provided environment in the future. | 0.190 | 0.112 | 0.920 | 0.153 |
| | | A15 | I want to spend more time using the provided environment. | 0.193 | 0.089 | 0.908 | 0.199 |
| | | A14 | The environment helps with my programming activities. | 0.552 | 0.237 | 0.564 | 0.301 |
| Positive Perceptions of Programming | | B02 | Programming is worth studying. | 0.281 | 0.198 | 0.113 | 0.862 |
| | | B01 | Programming helps create a better world. | 0.211 | 0.238 | 0.119 | 0.846 |
| | | B04 | Programming is relevant to the environment, technology, and society. | 0.179 | 0.138 | 0.136 | 0.841 |
| | | B03 | Programming will be useful even after I graduate school. | 0.270 | 0.179 | 0.085 | 0.814 |
| | | B07 | I want to know more about programming. | 0.275 | 0.191 | 0.141 | 0.790 |
| | | B05 | The programming class hours at school should be increased. | 0.282 | 0.193 | 0.179 | 0.782 |
| | | B06 | Programmers think and make decisions rationally. | 0.312 | 0.153 | 0.215 | 0.679 |

**Table 6.** Averages and standard deviations for each grade level.

| Factor | M (SD) | | | |
|---|---|---|---|---|
| | Middle School | High School | University | Total |
| Understanding of Programming Instructions | 4.04 (0.58) | 4.35 (0.74) | 4.63 (0.68) | 4.46 (0.71) |
| Usage Confidence | 3.80 (0.64) | 4.28 (0.75) | 4.53 (0.67) | 4.34 (0.73) |
| Usefulness | 4.18 (0.69) | 4.34 (0.70) | 4.16 (0.96) | 4.21 (0.86) |
| Positive Perceptions of Programming | 4.58 (0.43) | 4.70 (0.48) | 4.72 (0.57) | 4.69 (0.53) |

4.2.4. Regression Analysis

A multiple regression analysis was performed to determine the effects of the "programming learning environment" on "positive perceptions of programming" at each grade level. Stepwise selection was used to input the independent variables into the analysis, and SPSS for Windows (version 26.0) was used for regression analysis.

## 5. Results

The results of analyzing the effect of the "programming learning environment" factor on "positive perceptions of programming" are as follows: The "programming learning environment" factor includes "understanding of programming instructions, usage confidence, and usefulness". The analysis of variance results show that the usability factor had a statistically significant effect on "positive perceptions of programming". Table 7 shows the results of the detailed analysis of the factors that affected the "positive perceptions of programming".

**Table 7.** The analysis of variance summary of the correlation.

| ANOVA | | | | | |
|---|---|---|---|---|---|
| **Model** | **Sum of Squares (SS)** | **df** | **Mean Square (MS)** | **F** | |
| Regression | 16.144 | 2 | 8.072 | 53.133 | 0.000 |
| Residual | 18.990 | 125 | 0.152 | | |
| Total | 35.134 | 127 | | | |

Table 8 represents the analysis results. "understanding of programming instructions" had a statistically significant effect on "positive perceptions of programming" with a Beta value of 0.462 and a significance level of 0.5. The "usage confidence" factor was also statistically significant with a Beta value of 0.248 ($p < 0.05$). Therefore, it can be considered that using the programming environment and increasing the confidence made the programming perceptions more positive.

**Table 8.** The regression coefficient of the correlation.

| **Model** | **B** | **Std. Error** | **Beta** | **t** | **Sig.** |
|---|---|---|---|---|---|
| (Constant) | 2.393 | 0.225 | | 10.612 | 0.000 |
| Understanding of Programming Instructions | 0.342 | 0.082 | 0.462 | 4.170 | 0.000 |
| Usage Confidence | 0.178 | 0.080 | 0.248 | 2.233 | 0.027 |

The B-value was used to predict the programming perceptions based on the programming learning environment. The following regression equation was derived using the B (unstandardized coefficient) value of the three factors, and the perceptions of the programming value can be predicted based on "understanding of programming instructions" and "usage confidence".

(a) All Students $R_{eg}$ = 2.393 + 0.342 × (Understanding of Programming Instructions) + 0.178 × (Usage Confidence);
(b) Middle School Students $R_M$ = 0.426 × (Usefulness) + 2.804;
(c) High School Students $R_H$ = 0.280 × (Usefulness) + 0.251 × (Understanding of Programming Instructions) + 2.395;
(d) University Students $R_U$ = 0.375 × (Understanding of Programming Instructions) + 0.277 × (Usage Confidence) + 1.724.

For example, out of all students (a), the positive perceptions toward programming of those who answered 5 for "understanding of programming instructions" and 3 for "usage

confidence" was calculated as $R_{eg} = 2.393 + (0.342 \times 5) + (0.178) \times 3$, and their perception toward programming was 4.637.

The explanatory power of the "programming learning environment" factor with respect to "positive perceptions of programming" was 45.9%. Of this, "understanding of programming instructions" was 43.8% and "usage confidence" was 2.1%. When we examined results by school level, the explanatory power of "usefulness" on "positive perceptions of programming" was found to be 45.7% for middle school students. For high school students, "usefulness" was found to be 41.1%, whereas "understanding of programming instructions" was found to be 9.8%. For university students, "understanding of programming instructions" and "usage confidence" were found to be 52.0% and 2.7%, respectively.

## 6. Discussions

In the programming education for beginners, it is important for beginners to maintain a positive perception throughout their learning process. In other words, an increase in learning efficacy is expected to have a positive effect on motivation to continue learning [24,25]. Accordingly, hybrid-based programming environments studies have been conducted wherein text-based programming was performed in block-based programming environments [21,22]. Although this hybrid-based programming environment approach has demonstrated effectiveness, it has not been shown to provide specific factors that instill positive perceptions of programming. In this study, we utilized a block-based programming environment [5–9], as such environments have been shown to alleviate negative perceptions regarding typos and errors [10,11,17]. We also analyzed the factors associated with positive perceptions of programming. Based on the analysis, the discussion of the research results is as follows:

First, we must consider the fact that interest in and understanding of programming may vary with respect to school level, even in beginners. In the case of "understanding of programming instructions", middle school students were found to exhibit the lowest level of understanding (4.04), whereas university students displayed the highest level of understanding (4.63). Whereas middle school students could use the basic instructions presented in this study, university students successfully created unique programs according to these basic instructions. Furthermore, "understanding of programming instructions" was linked to "usage confidence", measured at 3.80 and 4.53 for middle school and university students, respectively, indicating that the development of programs relates to confidence. In contrast, "usefulness" of programming was similar between middle school (4.18) and university (4.16) students. These results are consistent with those of a previous study [19], which suggest that success in programming leads to satisfaction with the learning process itself. Likewise, the results obtained in [10,11,50] suggest that an insufficient understanding of programming instructions decreases overall satisfaction in the classroom. In other words, a sense of accomplishment may instill positive perceptions of programming in beginners. Therefore, it is necessary to stimulate a sense of confidence in students based on their understanding of instructions.

Second, the direction of the class should be set differently depending on the school level. The results of analyzing factors that affect "positive perceptions of programming" show that "usefulness" was a factor for middle school students, "usefulness" and "understanding of programming instructions" were factors for high school students, and "understanding of programming instructions" and "usage confidence" were factors for university students. In this study, "Usefulness" referred to whether the programming environment was assisting in programming. Therefore, it is necessary to build an environment that minimizes difficulties for middle and high school students. This is consistent with the results of studies that have reported that block-based programming environments cultivate learners' interest and motivate them to continue programming because these environments have no errors [51–53].

Unlike middle and high school students, the "usage confidence" factor affected "positive perceptions of programming" among university students. Confidence in programming

ability can strengthen positive perceptions among students. This is consistent with the results of studies that have reported that beginners lose confidence and interest when programming education is conducted solely in a text-based environment. However, they use commands confidently and are motivated to continue programming when text-based programming is performed in a block-based programming environment [54,55]. University students perform programming either because it is a requirement of their course or to gain employment; therefore, text-based programming will help them in a practical manner [56].

The factors that influence high school students' positive perceptions of programming are "usefulness" and "understanding of programming instructions". Between these factors, the "usefulness" also affected middle school students, and the "understanding of programming instructions" also affected university students. As high school students are in an intermediate stage of growth, a balanced approach should be utilized to teach programming to high school students by considering the learning dispositions of middle school and university students. In other words, it is important to consider simultaneously the aspect of university students experiencing reduced motivation due to the boredom of using error-free and easy-to-use block-based programming environments and the aspect of middle school students finding text-based programming classes challenging, given that they are beginners in programming.

To lower the barriers of introductory programming, classes must be tailored to the learning disposition of each school level.

## 7. Conclusions

With the increasing importance of programming education, many studies have been conducted to support beginners. Despite various studies on programming environmental factors, learning motivation, learning effectiveness, and satisfaction, the challenges faced by programming beginners remain an issue that needs to be addressed in programming education. Therefore, this study was conducted to identify factors associated with positive perceptions of programming in beginners. To achieve the research objectives, the school levels were categorized into middle school, high school, and university, and classes were conducted over a 14-week period. The results of the analysis of factors influencing positive perceptions of programming are as follows:

First, we found that factors associated with positive perceptions of programming varied according to school level. For middle school students, the "usefulness" of the programming environment, and for university students, "understanding of programming instructions" and "usage confidence" were the factors that influenced positive perceptions of programming. For high school students, the factors influencing "positive perceptions of programming" were found to have an intersection with both middle school and university students. In other words, the factors affecting "positive perceptions of programming" among high school students are "usefulness" and "understanding of programming instructions".

Second, for university students, "usage confidence" emerged as a significant factor, unlike middle school or high school students. These results are consistent with those of a previous study, which suggests that university students prioritize the outcome and practicality of programming, as programming may be necessary for their careers [56]. The findings of this study demonstrate that the programming environment and educational methods should vary with respect to the students' objectives and perceptions.

Based on these results, we propose the following directions for future research:

First, research is needed to correct the reasons for beginners failing in the process of programming according to their school level. For middle school students who need to understand basic commands, feedback should be provided that can resolve errors. For university students working toward employment, feedback must be provided on a higher difficulty level to stimulate their confidence. Moreover, because middle school or high school students may vary in terms of grade-level development stages and learning comprehension, it is necessary to elucidate the direction in which failures should be corrected in the programming process.

Second, there is a need for comparative research on different types of programming environments. There are hybrid studies that consider both text- and block-based programming languages. For instance, in some studies, block-based environments have been constructed for learning about text programming languages, and there have been studies that lower the barrier to entry to programming for beginners through the construction of hybrid-type environments. Despite these various studies, there is still a debate regarding appropriate environments for programming beginners at each school level. Thus, it is necessary to conduct research to compare traditional text programming language-based classes with classes in different types of block-based environments. Research on programming environments based on each school level will help to identify the characteristics of the environments that middle school, high school, and university students require to get started in programming and to conduct programming education that considers the developmental stages of students.

Third, there is a need for an environment that provides learners with various programming languages. It is necessary to provide environments for block-based text programming in not only Python but also C, Java, and other languages allowing users at different school levels the opportunity to choose a programming language that is suitable for their level.

This study is significant in that it demonstrates the need for different approaches to programming education, considering learning dispositions based on school levels, even for the same group of programming beginners.

**Author Contributions:** Conceptualization, Y.K. and W.L.; methodology, J.K. and Y.K.; software, Y.K.; validation, W.L. and J.K.; writing—original draft preparation, Y.K.; writing—review and editing, Y.K. and J.K.; supervision, J.K. and W.L. All authors have read and agreed to the published version of the manuscript.

**Funding:** This work was supported by the National Research Foundation of Korea (NRF) grant funded by the Korea government (MSIT) (No. NRF-2021R1A2C2013735).

**Institutional Review Board Statement:** Not applicable.

**Informed Consent Statement:** Informed consent was obtained from all subjects involved in the study.

**Data Availability Statement:** Not applicable.

**Conflicts of Interest:** The authors declare no conflict of interest.

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
