# Peer review of "Effect of Block-Based Python Programming Environment on Programming Learning"

_applsci, doi:10.3390/app131910898_

Round 1

Reviewer 1 Report

The provided article could be interesting because of these strengths:
1. The block-based approach reduces the cognitive complexity of text-based programming, making it more user-friendly and accessible for beginners.
2. Learners gain confidence in using programming commands within the block-based environment and a sense of assurance in their programming skills.
3. The block-based environment effectively supports university students' transition to text-based programming, demonstrating its applicability in higher education settings.
However, I found some weaknesses that should be discussed in the revised version:
1. The absence of a more fully presented comparison with traditional text-based instruction limits the study's ability to fully assess the approach's superiority (honestly, we can compare approaches on a similar group of students, and I see no problem doing it in your university).
2. The exclusive focus on the Python programming language restricts the research's external validity to other programming languages. Considering the growing diversity of programming languages, a broader examination of the block-based approach with various languages would enhance the study's applicability.
3. The study's duration covers only one semester, and the long-term effects of the block-based environment on participants' programming abilities remain unexplored.
4. The research does not explicitly explore other potential variables that may impact participants' perceptions of programming. Other psychological or cognitive aspects could have been relevant to consider.
5. The study's validity may be limited due to its only focus on Python (while the sample size and selection are comprehensive). The effects on learners using different programming languages remain unexplored.
6. Please provide some examples in Python (quizzes, exercises, etc.) for a more complex understanding of the methodology.

Author Response

We appreciate the reviewer’s valuable comments.

Our response is in the attached file. The revised portions from the paper are highlighted in red.

Thank you.

Reviewer 2 Report

The article investigates the impact of a block-based Python programming environment on programming education for beginners. The study conducted classes for students across various education levels to assess the effectiveness of this environment.

Positive Aspects:

The abstract provides a clear overview of the article's focus, which is the effect of block-based programming environments on programming learning, especially for beginners.

The study acknowledges the importance of programming education in the context of technological advancements and national competitiveness.

The article references various block-based programming environments like Scratch, Code.org, Blockly, and Pencil Code, highlighting their role in introducing beginners to programming and mitigating errors.

The research involves a substantial sample size of 128 middle school, high school, and university students, which adds credibility to the findings.

Areas for Improvement:

The article acknowledges a crucial aspect – the need for environments that support debugging and error resolution, which are integral parts of programming. However, it would be valuable to suggest more specific avenues for future research in this direction.

The conclusion mentions the need for environments that provide feedback and debugging support for beginners, but it doesn't delve into the potential ways such support could be integrated.

The article cites relevant studies to back up its claims, but it would have been helpful to include a few more of the most recent sources to provide the most up-to-date and comprehensive look at the topic.

Author Response

(The authors gave the same response as above.)

Reviewer 3 Report

It’s a great pleasure to review this article addressing a meaningful issue. Some comments are suggested as follows.

1. Because block-based and hybrid programming environments are the central theme of this study, the authors could summarize the current status in terms of major players and market share to give readers explicit background knowledge.

2. Similarly, the authors could present a literature review with comments on related studies to stress the differentiated advantages of this study.

3. It seems more appropriate to put “factor analysis” and “regression analysis” in the section “Method.”

4. It is essential for the authors to introduce sources of the 48 evaluation items. Besides, the authors could stress the need to develop new measurement scales.

5. Since background knowledge and requirements for different grade levels might be quite diverse, the explanations and implications of research findings need in-depth discussion.

5. Variables in Fig 1. need some explanations. The title of Fig. 4. seems not appropriate.

6. The discussions and conclusions could link stronger with research results with significant theoretical and piratical implications.

Author Response

(The authors gave the same response as above.)

Round 2

Reviewer 1 Report

The authors well addressed all my comments.

Author Response

I appreciate your review and comments.

Reviewer 3 Report

I appreciate the authors’ great efforts in revising their manuscript. Nevertheless, it is expected that the presentation of their research findings could be more organized and focused on the main themes. Some comments for the conclusion and discussion are suggested as follows.

1. The discussions seem to focus on explaining four regression equations. But before this, readers would expect more explicit ideas concerning factors producing different results. They want to see the forest before seeing the trees.

2. The conclusions could have stronger links with significant research findings and address more implications to reflect on research objectives.

3. Since some statements are confusing, for example, in Line 394, the author mentioned, “…using block-based programming has limitations.” Still, in Line .434, the authors stressed that “… with block-based programming tools performed significantly better.“ In their conclusions, the authors could have more assertive and consistent standpoints on the roles of block-based programming.

Author Response

I appreciate your review and comments.

We have revised the paper in accordance with your comments. The modified sentences are highlighted in red. Thank you.
